# NETWORK-BASED ACTIVE INFERENCE FOR ADAPTIVE AND COST-EFFICIENT REAL-WORLD APPLICATIONS: PV PANEL INSPECTION

## ABSTRACT

This paper introduces Network-based Active Inference (NetAIF), a novel framework that integrates random attractor dynamics and the Free Energy Principle (FEP) to improve trajectory generation and control in robotics. NetAIF optimizes the intrinsic dynamics of neural networks, enabling robots to quickly adapt to dynamic and complex real-world environments with minimal computational resources and without the need for extensive pre-training. Unlike traditional learning methods that rely on large datasets and prolonged training periods, NetAIF offers a more efficient alternative.

In real-world scenarios, such as Photovoltaic (PV) panel inspections, NetAIF demonstrates its ability to execute dynamic tasks with both high efficiency and robustness. The system excels in unpredictable environments while maintaining a low computational footprint. These capabilities make NetAIF a promising solution for industrial applications, offering cost-effective, adaptive robotic systems that can reduce operational expenses and enhance performance, particularly in sectors like energy, where adaptability and precision are crucial.

## 1 INTRODUCTION

### 1.1 OVERCOMING AUTOMATION CHALLENGES WITH ADVANCED LEARNING METHODS

The World Energy Employment 2023 report by the IEA highlights a significant shift towards clean energy jobs, which now surpass fossil fuel employment, driven by a 40% rise in clean energy investment over the past two years. Despite economic and geopolitical challenges, the energy sector has seen growth in employment, particularly in solar PV, wind, EVs, and battery manufacturing. However, a shortage of skilled labor remains a key challenge, underscoring the need for targeted training and policy support to develop a workforce suited for the energy transition (IEA, 2023).

In response to these labor challenges, automation is playing an increasingly critical role in advancing the clean energy sector. Robotics, in particular, offers a promising solution to enhance operational efficiency and safety. However, to maximize the potential of robotics in complex and dynamic environments, sophisticated learning methods are required. One such approach, Deep Reinforcement Learning (DRL), has emerged as a leading candidate for enabling autonomous robotic systems in tasks like control, manipulation, and decision-making. Yet, despite its potential, DRL faces notable barriers to widespread adoption in the energy sector.

### 1.2 DEEP REINFORCEMENT LEARNING (DRL)

DRL combines the decision-making power of reinforcement learning (RL) with the pattern recognition capabilities of deep learning (DL). This allows robots to learn and adapt through trial and error, improving performance over time. DRL is increasingly explored for enabling autonomy in control and manipulation tasks in real-world environments by training agents to recognize complex patterns in data and make informed decisions.

However, DRL requires large amounts of data and time for agent training, as well as expert-designed reward functions to guide learning. Creating these reward functions demands substantial knowledge

and engineering resources, as they must accurately capture desired outcomes, agent actions, and constraints. Poorly defined reward functions can lead to suboptimal or unsafe behavior (Sutton & Barto, 2020). Thus, while powerful, DRL may not always be the most practical or cost-effective approach for every application.

### 1.3 AIF as a Next Generation Learning Method

Active Inference (AIF) is a groundbreaking framework in neuroscience, offering a unified approach to understanding adaptive systems, including brain functions, and is gaining traction in fields like machine learning and robotics (Friston et al., 2006; Parr, 2019; Millidge, 2020; Lanillos et al., 2021). In robotics, AIF is reshaping control and learning by minimizing surprise rather than relying on reward-based mechanisms like DRL. Unlike DRL, which requires fixed environments, AIF utilizes a dynamic generative model, continuously adapting to changing surroundings through a feedback loop of prediction, perception, and action. This approach addresses the exploration-exploitation dilemma more fluidly by incorporating uncertainty directly into decision-making.

While AIF holds significant promise for creating adaptive robotic systems, its real-world deployment faces challenges due to the complexity of model design and high computational demands (Lanillos et al., 2021). Nonetheless, its potential to enhance flexibility, durability, and adaptability makes it a powerful alternative to traditional DRL techniques

### 1.4 Network-based Active Inference (NetAIF)

To overcome the limitations of both DRL and traditional AIF approaches, we propose Network-based Active Inference (NetAIF), a novel framework that leverages network dynamics to simplify trajectory calculations and enhance efficiency. Rooted in key AIF principles such as entropy and surprise minimization, NetAIF builds on the Free Energy Principle (FEP), which posits that systems self-organize by minimizing surprisal or prediction error. By harnessing the inherent dynamics of a network, NetAIF computes trajectories more efficiently than traditional AIF methods, reducing the need for complex mathematical models while enabling agents to adapt to dynamic environments in real-time. This streamlined approach makes NetAIF highly suitable for real-world robotic applications, offering significant improvements in both speed and computational cost.

## 2 NETWORK-BASED ACTIVE INFERENCE

### 2.1 Notable Characteristics

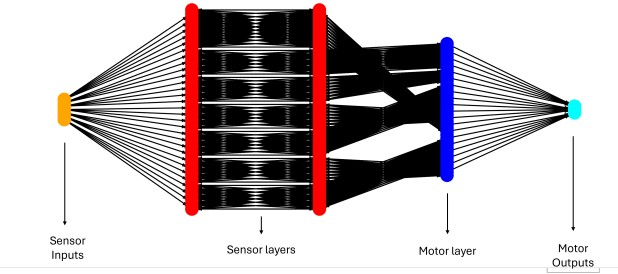

Figure 1: NetAIF network diagram for target-tracking task: parameters that determine the network structure such as number of layers, strides were determined through hyper parameter search

NetAIF's key innovation lies in its explicit feedback loops between hidden layers, which deliberately induce controlled instabilities to explore the state space more thoroughly (Brown, 2021)(Refer to Figs. 1 and 2). Unlike Recurrent Neural Networks (RNNs), where feedback is implicit (Mienye et al., 2024), NetAIF actively manipulates network dynamics to push the system into unstable regions. These feedback loops enhance oscillatory patterns, similar to neuron firing sequences, that

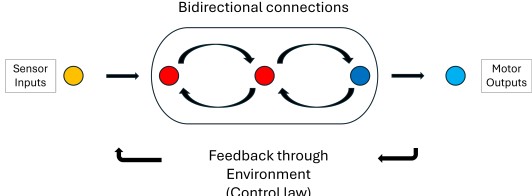

Figure 2: bidirectional connection in hidden layers: the schematic diagram shows how the instability is induced within the hidden layer and how such instability is controlled via the external control law through feedback

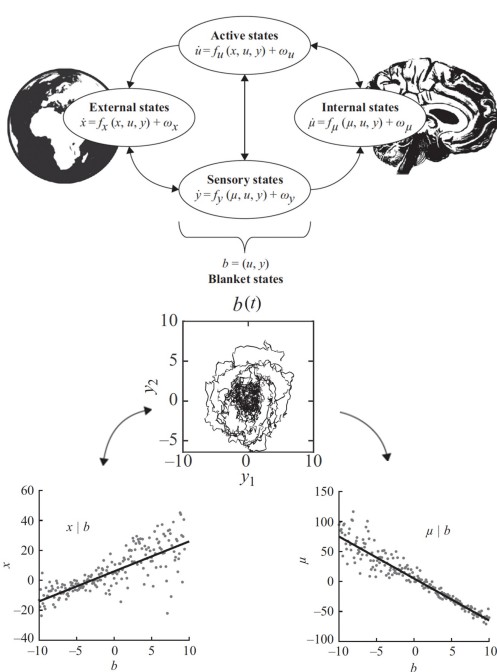

Figure 3: AIF brain and world - External states (world) are mirrored by internal states (brain). The active and sensory states (blanket states) couple external to internal states-rendering the system open. The (far from equilibrium steady-state) dynamics of each state is described with stochastic differential equations (*w* is a stochastic fluctuation). The images were adapted and modified from Parr et al. (2022)

persist even after training. This random bursts of node activity can be observed in the supplementary video, further highlighting the parallels with brain function. The introduction of these instabilities enables the system to maintain dynamic behaviors, known as itinerant (wandering) dynamics (Kaneko & Tsuda, 2003; Friston & Ao, 2012), allowing it to continuously adapt to changing environments.

NetAIF operates within the framework of Active Inference, where a system interacts with its environment through blanket states. Blanket states consist of sensory states, which gather external information, and active states, which influence the environment as shown in Fig.3. This dynamic interaction forms the core of the system's ability to operate in a Non-Equilibrium Steady State (NESS). In NESS, the system is never fully at rest but continuously adapts to changing inputs from the environment, minimizing prediction errors in real time. The feedback between sensory and active states ensures that the system remains stable yet flexible, adjusting its actions and beliefs to maintain optimal performance even in uncertain or complex environments. This aligns with Bayesian inference principles, as NetAIF constantly updates its beliefs in response to new sensory inputs and envi-

ronmental changes, enhancing its ability to navigate complex environments and discover optimal trajectories.

NetAIF also replaces traditional activation functions with a discrete weight-assigning mechanism, designed to reset node weights and maintain NESS. By leveraging the constant interaction between sensory and active states, NetAIF remains in a state of continuous exploration, avoiding local minima and ensuring that it adapts dynamically to new challenges. This stochastic function enhances the network's ability to explore different states, preventing it from being trapped in local optima.

Additionally, NetAIF integrates learning and control, guiding motor outputs with clear task-specific control laws. These laws break tasks down into sub-goals, such as aligning objects, allowing even non-experts to define behaviors without deep control theory knowledge. For instance, in a valve manipulation task, control instructions guide the network to minimize errors by aligning the vector of the valve's position with the one of the end effector. This ensures precise orientation and movement, making the system more intuitive and effective for real-world applications. This user-friendly approach facilitates seamless integration of learning and control.

An effective way to understand this is through an analogy: the control (vector) law acts like a road, providing a set of boundaries, while the random attractor serves as the driver, navigating the road to find the optimal path in real-time. Just as a driver adjusts their route based on obstacles and traffic while staying on the road, the random attractor dynamically explores within the constraints set by the control law, ensuring the robot adapts to changing conditions while maintaining the most efficient trajectory. This approach allows for greater flexibility and precision in the robot's movement.

---

**Algorithm 1** Main loop of the NetAIF model

---

1: **Initialize** all model parameters and weights
2: **while** system is running **do**
3:     Prediction_Error = $Desired\_State - Current\_State$
4:     Input_signals = $Prediction\_Error$
5:     **for** each weight $w$ in all weights **do**
6:         **if** magnitude of associated signal $>$ threshold **then**
7:             Set $w = new\_weight\_value()$
8:         **end if**
9:     **end for**
10:     Input_to_hidden = $Input\_signals \times W\_input\_hidden$
11:     Feedback = $Hidden\_signals\_prev \times W\_hidden\_hidden$
12:     Hidden_signals = $Input\_to\_hidden + Feedback$
13:     Hidden_signals_prev = $Hidden\_signals$
14:     Outputs = $Hidden\_signals \times W\_hidden\_output$
15:     Motor_Commands = $Outputs$
16:     Send motor commands to actuators
17: **end while**

---

The core of the NetAIF framework is outlined in Algorithm 1. Each cycle calculates the prediction error between current and desired states, which updates network weights dynamically. If a signal exceeds a set threshold, its weight is reset to ensure stability. Feedback loops in the hidden layers facilitate adaptive behavior and robust trajectory generation. Motor commands are derived from the hidden layers and sent to the actuators, enabling real-time adjustments. This continuous feedback allows NetAIF to quickly adapt to changing environments, making it ideal for dynamic tasks like PV panel inspection.

## 2.2 THE RANDOM ATTRACTOR

To represent the NESS behavior in NetAIF, Random Dynamical Systems (RDS) are employed, providing a framework to understand complex systems driven by stochastic processes. In particular, random pullback attractors (Caraballo & Han, 2016), also known as stochastic basins of attraction, describe how NetAIF's state evolves over time in response to environmental uncertainty. Expressed as $\varphi(t, \omega, x)$, where $t$ is time, $\omega$ represents randomness, and $x$ is the state variable, these attractors characterize regions in the state space where the system tends to settle. The random attractor $\mathcal{A}(\omega)$ pulls trajectories towards it, ensuring that NetAIF remains adaptive and stable within its NESS framework, despite external randomness.

This is formalized by:
$$\lim_{t \to \infty} \text{dist}\left(\varphi(t, \theta_{-t}\omega, B), \mathcal{A}(\omega)\right) = 0$$

where $\varphi(t, \theta_{-t}\omega, B)$ represents the state of the system at time $t$, $\theta_{-t}\omega$ is the time-shifted random noise, where $\theta$ is a shift operator that moves the noise backward in time by $t$ units. This term captures the idea that the noise affecting the system at time $t$ is related to the noise that occurred in the past. $B$ is a bounded set of initial conditions, and $\text{dist}(X, Y)$ denotes the distance between sets $X$ and $Y$.

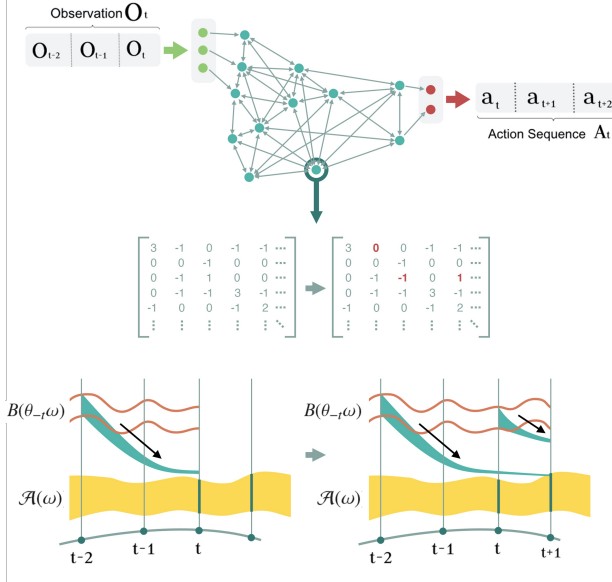

Figure 4: Abstract representation of a random pullback attractor, $\mathcal{A}$, and the random set, $B$. While the weights of the network are updated randomly (shown in matrix format), a flow from the random set emerges and gets attracted to the attractor.

This convergence process can be understood as a stochastic diffusion in parameter space, driven by increasing the amplitude of random fluctuations on parameters (e.g., connection weights) in regions of high free energy. As the system approaches free energy minima, these random fluctuations are attenuated, resulting in a more stable and precise arm trajectory. Such system dynamics can be described by a stochastic differential equation (SDE) in the form of a Langevin equation (Karl, 2019):

$$dx = -\nabla F(x)\, dt + \sqrt{2\Gamma}\, dW$$

where $x$ represents the system's parameters, $F(x)$ is the free energy landscape, $\Gamma$ is the diffusion coefficient, and $W$ is a Wiener process. This equation captures the interplay between the deterministic drift towards free energy minima and the stochastic exploration of the parameter space, which ultimately shapes the arm's trajectory.

It is worth noting that the optimization process in NetAIF is inherently local because free energy is an extensive quantity, meaning that the system's total free energy is the sum of the free energies of its individual components. The variational free energy, which approximates the true free energy, is calculated using local prediction errors. Some predictions are clamped with high precision, fixed, or strongly influenced by the desired outcomes, defining the attracting set, which represents the desired sensor inputs or the target state of the system. Minimizing variational free energy by reducing local prediction errors guides the network model towards the attracting set.

This local optimization process enables the system to efficiently navigate the free energy landscape without requiring global computations or information propagation across the entire network. By iteratively updating its local components based on prediction errors and external control laws, the system converges towards the desired states.

The roots of this learning scheme can be traced back to early formulations of self-organization in cybernetics (Ashby, 1947) (Ashby, 1956) and are connected to stochastic thermodynamics (Ao, 2008) (Seifert, 2012). These connections highlight the consistency of the design principle with the fundamental concepts underlying the FEP. This principle drives the network model to minimize pre-

diction errors, guiding the entire network towards a stable regime, resulting in smooth and efficient arm movements.

# 3 APPLICATION: PV PANEL INSPECTION

## 3.1 PROJECT SCOPE

PV farms, covering extensive areas spanning numerous hectares, traditionally rely on the keen eyes of professional inspectors to identify damages or issues on the panels. This method of manual inspection, while thorough, is both time-intensive and laborious given the vastness of the installations. The complexity of the task is exacerbated by the dynamic nature of PV farms, which face continual changes due to weather patterns, the undulating terrain, the encroachment of wildlife, and the unpredictable intrusion of vegetation. These factors contribute to noisy data and unforeseen challenges for any automated systems (such as robotic solutions) that might be employed to streamline the process. Furthermore, the variability in panel aging, weather fluctuations, and diverse panel types necessitate robot controllers with an increased level of adaptability and flexibility.

To address these challenges, we have implemented an inspection system using a robotic arm equipped with sensors designed to detect PV panel defects. This system had to meticulously extend to a pre-determined distance from a panel, position its sensors to be perpendicular to the panel's surface, and systematically survey the panel sections for any signs of deterioration or damage.

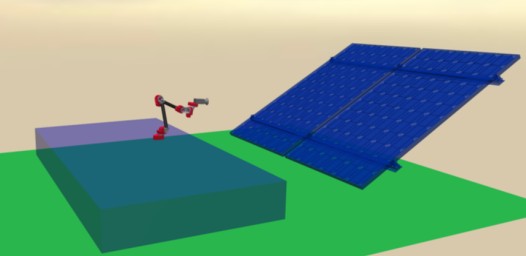

Figure 5: PV panel inspection simulation environment (Mujoco)

## 3.2 SPEED AND DISCERNMENT

To simulate complex real-world conditions, we established a test environment featuring the HEBI 6-DoF SEA (Series Elastic Actuator) robotic arm affixed to a mobile base (Refer to Fig. 5). The combination of the arm's compliant actuators, which provide additional degrees of freedom, and the unpredictable movement of the mobile platform, effectively replicates the difficulties encountered in real-life operational settings. Additionally, the HEBI arm's modular design was particularly advantageous for testing the NetAIF model's flexibility; the arm can be readily reconfigured into 4, 5, or 6-DoF configurations. This feature is vital in practical applications where an arm might experience actuator failure and still need to function effectively. In this setup, the objective was for the camera-mounted arm to navigate to a predetermined location—the midpoint of a PV panel—and adjust its orientation to be perpendicular to the panel's surface. The NetAIF controller's task was to preserve this precise position and orientation during the scanning process.

The NetAIF model successfully acquires tasks quickly, and does so with notable accuracy, as outlined in Table 1. It is worth pointing out that relatively big variation in distance measurements can be attributed to the positioning of the camera-equipped arm far from the panel during inspection tasks. Moreover, for optimal inspection outcomes, the camera's orientation angle holds more significance than its exact alignment with the panel's center.

## 3.3 ROBUSTNESS AND ADAPTABILITY

To assess the resilience and flexibility of NetAIF, we progressively expanded the robot arm's movement until a noticeable discrepancy occurred. In this context, a noticeable discrepancy is defined

Table 1: PV panel inspection performance metric with stationary base

| Model Performance (with stationary base) | | | |
|---|---|---|---|
| | 4DOF | 5DOF | 6DOF |
| Distance Error | 3.2 mm | 16 mm | 14 mm |
| Yaw Error | 0.15° | 0.065° | 0.070° |
| Pitch Error | 0.069° | 0.086° | 0.089° |
| Roll Error | 0.016° | 0.051° | 0.023° |

as a 5 cm divergence from the PV panel's center and a 5-degree deviation from the perpendicular alignment. As indicated in Table 2, each configuration of the HEBI arm successfully tolerated a substantial level of random movements and orientations.

Table 2: PV panel inspection performance metric with moving base - The tolerance was measured for all axes movements, showing the minimum value that triggered a significant divergence (i.e., the higher the tolerated amount, the more robust)

| Model Performance (with moving base) | | | |
|---|---|---|---|
| | 4DOF | 5DOF | 6DOF |
| Random Motion Tolerated | 30 cm/sec | 50 cm/sec | 50 cm/sec |
| Random Orientation Tolerated | 20°/sec | 30°/sec | 45°/sec |

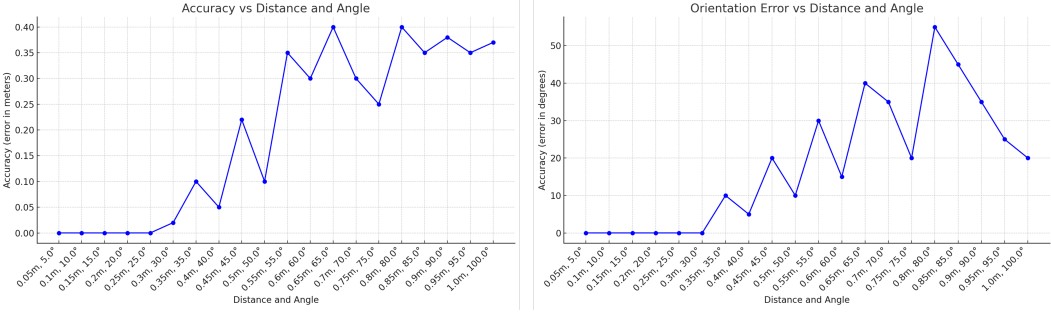

Figure 6: PV panel inspection with moving base - Position and orientation deviation across all axes as the base simultaneously experiences random movements and tilting

Fig. 6 reinforces the aforementioned findings. Displayed within these graphs is the aggregated maximum displacement or orientation across all axes for the 6DoF arm, captured while the base undergoes concurrent random movements and tilts.

## 4 EXPERIMENTS

We conducted two key experiments with the physical Lite6 6-DoF arm from UFactory, operating at 100 Hz: a pose-matching task, which served as a benchmark, and a target-tracking task related to PV panel inspection. In the pose-matching test (Fig. 7), the joint pose was directly fed into the system, and the attractor calculated waypoints for a smooth and efficient trajectory to move the robot to the specified pose. The control law was simple, designed to match the current joint position with the desired one. The arm smoothly and efficiently reached the predetermined position, showcasing the effectiveness of using attractor dynamics for trajectory generation without explicit path planning algorithms.

For the target-tracking task, the robotic arm successfully learned to follow an AprilTag detected by a RealSense D455 camera, with accuracy enhanced by a Kalman filter Kam et al. (2018). Reference vectors were used to align the robot's roll, pitch, and yaw orientations with the moving target. Notably, the arm was able to track the marker in real-time without any need for pre-training.

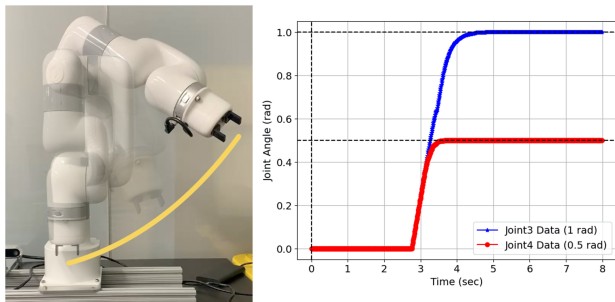

Figure 7: Network Output Signal for Pose Matching Task

The swift and efficient performance of the NetAIF model can be attributed to its FEP-guided path generation, combined with random attractor dynamics. As illustrated in Fig. 8, these random attractor dynamics replace conventional motion planning components. Unlike some of the traditional methods, where the entire trajectory is pre-calculated or trained, NetAIF generates the trajectory in real-time by continuously feeding sensor data to the random attractor, allowing for more flexible and adaptive motion planning.

Table 3 presents the performance metrics for the NetAIF model, evaluated on an 8-core Intel Core i9 (I9-9880H) 2.4 GHz processor without GPU support. The network's update cycle was approximately 7ms, as detailed in Table 4, resulting in a remarkably short training time of just about 8 seconds for the target tracking task. Once the network is trained, the resulting trajectory values become smoother with relatively small random fluctuations. This smoothness reflects the efficiency of the network's attractor dynamics, which generate real-time adjustments based on sensor data, allowing for precise tracking without requiring pre-calculated trajectories. Additionally, the model's stored weight values improve deployment flexibility, making it easily transferable and deployable across different systems. This portability ensures that similar tasks can be executed efficiently without requiring retraining, providing a key advantage—especially when the network is scaled up to handle more complex tasks.

Table 3: NetAIF Model Metrics

| Metric | Pose-Matching | Target-Tracking |
|---|---|---|
| Network Size (No. of Nodes) | 132 | 176 |
| Network Size (No. of Connections) | 1212 | 1616 |
| Network Size (No. of Bytes) | 10224 | 13632 |
| No. of Iterations to Convergence | 955 | 1230 |

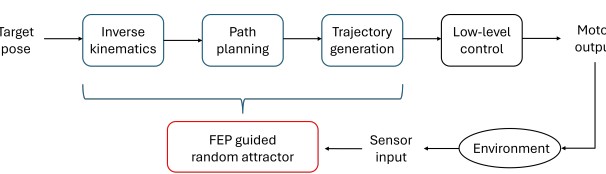

Figure 8: Motion planning process

Fig. 9 shows two key visualizations that offer insights into the movement of a robot's joints and its end-effector trajectory. The left plot shows the evolution of joint positions over time for six joints. The right plot depicts the 3D trajectory of the April tag along with the end-effector, revealing a non-linear and intricate path with multiple loops and clusters.

Fig. 10 provides a cross-correlation analysis between a marker's position in the X, Y, and Z directions and six robot joints, revealing insights into how different joints influence the marker's

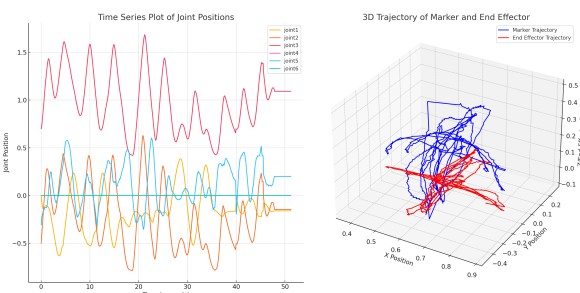

Figure 9: Joint and marker positions over time

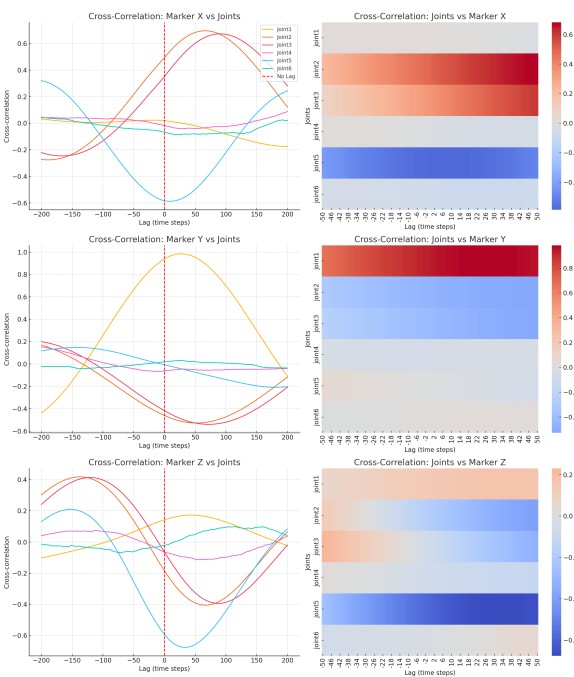

Figure 10: Time-lagged cross-correlations

movements over time. The analysis shows that certain joints lead or lag in their contributions to the marker's trajectory, highlighting the coordinated nature of the robot's movement, which is generated by the network's attractor dynamics. For instance, joints 2 and 5 show strong, delayed correlations with the marker's X position, suggesting that they are key contributors to larger, slower movements, responding after other joints have initiated motion. In contrast, joint 1 shows a stronger and more immediate influence on the marker's Y direction, likely because it controls base-level adjustments in the robot's workspace. The Z-axis motion involves more complex interactions, with joints 2 and 3 leading in correlation, suggesting they play a pivotal role in vertical positioning and correction. These leading and lagging behaviors arise due to the robot's kinematics—joints located closer to the base (like joint 1) may initiate broader movements, while those closer to the end-effector (like joint 5) respond later to fine-tune the motion or compensate for inertia. This reflects the coordinated effort between joints, generated by the network, to achieve precise, controlled movements, where some joints lead by initiating directional changes and others follow to stabilize or refine the movement.

The total motion planning time for a target-tracking task involving real-time visual processing is summarized in Table 4 and Fig. 11, with an average planning time of 6.7 milliseconds. This demonstrates the model's remarkable efficiency, especially considering the frequent need for replanning

due to environmental changes and moving targets. In comparison, algorithms such as PRM and Hybrid RRT-PRM can take up to 482 milliseconds for planning under similar conditions, largely due to the computational overhead involved in path updates (Jermyn, 2021). Likewise, UAV-based research with visual processing reports planning times ranging from 50 to 500 milliseconds in dynamic environments (Cui et al., 2022). Although the NetAIF model has a large standard deviation of 16.16 milliseconds, reflecting variability from factors like fluctuating frame rates and environmental dynamics, it still maintains an impressive mean of 6.7 milliseconds. This efficiency, even with frequent replanning, underscores the system's exceptional capability to handle complex, dynamic tasks with minimal computational delay.

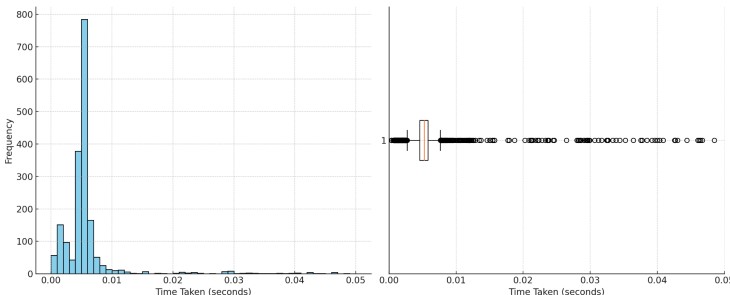

Figure 11: Total motion planning time

Table 4: Summary of time taken to generate values by the network

| Statistic | Value (milliseconds) |
| --- | --- |
| Mean time | 6.7 |
| Standard deviation | 16.16 |
| Median time (50th percentile) | 5.23 |
| 25th percentile | 4.56 |
| 75th percentile | 5.80 |

## 5 CONCLUSIONS

The Network-based Active Inference (NetAIF) model offers a novel, efficient approach to real-time adaptive intelligence in robotics by leveraging random attractor dynamics and the Free Energy Principle (FEP) to enable robots to adapt to unpredictable environments without extensive pre-training or significant computational resources. Its real-time feedback processing ensures precise control and flexible adaptation, making it ideal for cost-sensitive industries like energy, where adaptability and precision are critical. Unlike Deep Reinforcement Learning (DRL), which requires substantial training and computational power, NetAIF provides a computationally efficient, cost-effective solution for tasks such as inspections and maintenance. For a comparison with DRL methods, see the companion paper (Anonymous, 2024).

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
