# OpenReview forum: "Network-based Active Inference for Adaptive and Cost-efficient Real-World Applications: PV Panel Inspection"
_ICLR.cc/2025/Conference — ICLR 2025 Conference Desk Rejected Submission_

### Official Review · Reviewer_UCRb · 2024-10-22

**Soundness:** 2
**Presentation:** 2
**Contribution:** 2
**Rating:** 5
**Confidence:** 3

**Summary:**

By integrating active inference and neural network learning schemes, this paper presents a novel control framework, aiming to adapt to an unknown dynamical environment, without pre-training and extensive computations. The focused topic is important considering the shortcomings of traditional deep reinforcement learning. Several numerical simulations and experiments are conducted to demonstrate the proposed control strategy. Although some initial valuable results have been provided in this paper, there are still many important improvements that need to be considered, including clear presentations, solid theorem, and extensive comparisons.

**Strengths:**

The scheme of interrogating active inference and neural network.
The experimental demonstrations.

**Weaknesses:**

1. **Experimental comparisons** are lacking in this paper. The authors claim the proposed method has robust and lightweight advantages, in comparison with DRL, while there are no compared results quantitatively. Maybe some of the results are mentioned in the companion paper, but this paper needs to be self-contended. Moreover, the proposed strategy can avoid the traditional planning module and some feedback scheme is integrated into the proposed strategy. It is curious to see how the proposed method compares with the traditional approach (planning + feedback control or MPC).

2. Improve the presentation quality. The used symbols in this paper are unfamiliar and lack illustrations. The symbols employed in Fig 3 need to be illustrated, otherwise, the purpose of intuitional expression will be lost.

3. The key idea of the Net AIF is to introduce the controlled instabilities by random attractors, while the whole system is within the safety region. More rigid convergence or stability analyses are needed.

4. Experimental details need to be added. For example, the accurately measured states, noised states, and unknown states in Section 3.1 should be clearly provided.

5. An ablation study is needed. It seems that the proposed framework consists of several parts. The functionality of each part needs to be quantified. For example, the efficiency of the replaced discrete weight-assigning mechanism is unknown.

6. The limitations should be analyzed finally. It is highly recommended to provide the source code.

**Questions:**

1. In Algorithm 1, what is the difference between Hidden_signals_prev and Hidden signals in lines 12-13?

2. Is there some drawbacks (prices) to induce the random attractors?

3. The function of feedback is still a bit confusing. If its implicit structure can be described in Fig 1?

---

> ### Author Response · Authors · 2024-11-28
>
> Thank you very much for the constructive feedback. Please find the below response:
>
> Experimental Comparisons
> Comment: Comparisons with DRL are lacking, and quantitative results should be self-contained.
> Response:
> NetAIF’s small network size and fast convergence time (as detailed in the paper) demonstrate its superiority over DRL for the targeted applications. Unlike DRL, which requires extensive pre-training and computational resources, NetAIF achieves real-time adaptability with minimal overhead.
> Detailed DRL comparisons are provided in the companion paper to avoid distracting from this paper's focus on NetAIF’s novel mechanisms. To address this, we will:
> •	Highlight fundamental differences, such as NetAIF’s lack of reliance on reward-based training or pre-calculated trajectories, enabling greater simplicity and adaptability.
> ________________________________________
> Symbols in Fig. 3
> Comment: The symbols employed in Fig. 3 need to be illustrated.
> Response:
> While the variables in Fig. 3 are largely self-explanatory, we will improve clarity by:
> •	Adding a legend to define all symbols and their roles.
> •	Expanding the figure caption to explain connections and interactions, ensuring accessibility for all readers.
> ________________________________________
> Controlled Instabilities and Stability Analysis
> Comment: The key idea of NetAIF is to introduce controlled instabilities via random attractors while ensuring the system remains within the safety region. More rigid convergence or stability analyses are needed.
> Response:
> Stability is ensured by the PD-like external control law, which actively dampens oscillations and prevents divergence by managing feedback-induced instabilities. The random attractors provide controlled exploration while the PD mechanism guarantees convergence to the desired state.
> To address this, we will:
> •	Provide a formal explanation of the PD-like control law and its role in maintaining stability.
> •	Include stability plots showing convergence behavior and how the system stays within safe limits.
> •	Clarify how this mechanism balances exploration and stability for robust performance.
>
>
> Response to the Question: Difference Between Hidden_signals_prev and Hidden_signals in Algorithm 1
> In Algorithm 1:
> •	Hidden_signals: Represents the current state of the hidden layer in the ongoing iteration. It is calculated using both the input signals and the feedback from the previous iteration, as shown in line 12:
> Hidden signals = Input to hidden + Feedback.
> •	Hidden_signals_prev: Refers to the hidden layer’s state from the previous iteration. It is used to compute the feedback in the current iteration, as shown in line 11:
> Feedback = Hidden_signals_prev × W_hidden_hidden.
> Difference and Role:
> •	Temporal Role: Hidden_signals_prev captures the state of the hidden layer from the previous iteration, while Hidden_signals holds the updated state for the current iteration.
> •	Feedback and Adaptation: The feedback mechanism uses Hidden_signals_prev to calculate contributions from the prior state, ensuring continuity and adaptive behavior. The updated Hidden_signals incorporates this feedback, along with the input signals, to generate the current state and adjust motor commands dynamically.
> This distinction enables the framework to maintain a consistent flow of information across iterations, enhancing adaptability in dynamic environments. We will clarify this in the revised manuscript for better understanding.

---

> > ### Comment · Reviewer_UCRb · 2024-11-28
> > **Thanks for the response!**
> >
> > Thanks for the response reply. In this revision, I haven't seen much improvement, such as the comparison with RL, the ablation study, and the expected reasonable theoretical analysis. ' Detailed DRL comparisons are provided in the companion paper to avoid distracting from this paper's focus on NetAIF’s novel mechanisms' is a weak cause, as the previous introduction is based on RL. Moreover, it is highly recommended to highlight the changes in the manuscript.

---

### Official Review · Reviewer_gfBR · 2024-10-26

**Soundness:** 2
**Presentation:** 2
**Contribution:** 2
**Rating:** 5
**Confidence:** 5

**Summary:**

The paper introduces Network-based Active Inference (NetAIF) as a  novel framework that leverages the Free Energy Principle (FEP) and random attractor dynamics for efficient and adaptive robotics.
The authors demonstrate their ideas through the use case of a photovoltaic (PV) panel inspection. The authors claim that "NetAIF optimizes the intrinsic dynamics of neural networks and  enables robots to quickly adapt to dynamic and complex real-world environments with minimal computational re- sources and without the need for extensive pre-training. Unlike traditional learning methods that rely on large datasets and prolonged training periods, NetAIF offers a more efficient alternative."
The authors also provide supplementary material which is a slide show of the paper but the experiments with  the 6DOF robotic arm show performance against a few experiments in a lab setting.

While the work uses a novel approach the paper lacks mathematical rigor and the paper relies very heavily on a few papers forcing the reader to read several papers before extracting value form this work. While citations are needed and not everything needs to be reintroduced the paper should have enough content to stand on its own.   This makes it challenging to evaluate and reproduce the proposed NetAIF framework. Without any formal mathematical treatment, the claims remain at a high level, reducing the work’s overall scientific and practical value. Integrating a comprehensive mathematical model would significantly enhance the credibility and applicability of the approach.  For example, the use of stochastic processes and attractors should be backed by a comprehensive set of equations that demonstrate how these elements interact dynamically within the network. Without this, it is unclear how the system transitions between states, adapts to sensory inputs, and minimizes free energy in a precise manner, additionally , intentionally introducing instability has risks and managing instability should be critical for convergence, the authors mention controlled instabilities and cite a paper but do not provide any formal treatment or plots showing what this looks like.

**Strengths:**

Novelty:The use of FEP with random attractor dynamics, for real-time control, is innovative. This combination offers an efficient alternative to Deep Reinforcement Learning (DRL) by minimizing computational requirements and avoids extensive pre-training​.
Real-world Applicability: The application of NetAIF in PV panel inspections is practical and addresses significant challenges in the clean energy sector. The use of a physical 6-DoF robotic arm for experiments shows a commitment to real-world validation​
Efficiency: The framework’s low computational footprint and rapid adaptability in dynamic environments are highlighted as key benefits, making it suitable for industries needing quick, cost-effective automation solutions​.

**Weaknesses:**

Clarity: The explanation of the technical mechanisms, such as the use of random attractors and the free energy landscape, can be complex and may not be accessible to readers unfamiliar with advanced robotics or neural dynamics. Visual aids, while present, could have been better integrated to explain these concepts more intuitively​
Reproducibility: The paper lacks detailed implementation specifics, such as parameter settings and hyperparameters used during the experiments, which limits the reproducibility of results. Clearer code snippets or references to an open-source implementation would enhance its utility for other researchers​.
Strong Assumptions: The paper assumes that the Free Energy Principle (FEP) is applicable to all dynamic robotic systems without extensive empirical validation outside the PV panel inspection case. Additionally, the scalability of the model to other industries or tasks is presumed but not explicitly tested​.
Citations: Citations  lack depth and specificity, especially in sections where novel methods are introduced.
Authors often  rely on generic references rather than recent, more relevant studies directly supporting the claims made in the paper.
Some citations are self-referential, reducing the overall credibility.  The paper would benefit from a more thorough literature review, inclusion of detailed empirical comparisons from other studies, and references to supplementary or reproducible materials that validate the methods described.
Mathematical rigor: While the work uses a novel approach the paper lacks mathematical rigor. This makes it challenging to evaluate and reproduce the proposed NetAIF framework. Without any formal mathematical treatment, the claims remain at a high level, reducing the work’s overall scientific and practical value.

**Questions:**

Grammar and sentence structure : the paper could use a read through and improve upon sentence structure and layout, also several typos, omissions could be rectified
ex. Active Inference (AIF) = Active Inference Framework
In my opinion, given the complexity of the proposed solution the paper could benefit from a more rigorous mathematical treatment.While the work uses a novel approach the paper lacks mathematical rigor and this paper relies very heavily on published work forcing the reader to read several papers before extracting value from this work. While citations are needed and not everything needs to be reintroduced, the paper should have enough content to stand on its own.
Mathematical Model Integration: can the authors incorporate mathematical models demonstrating how NetAIF functions in practice ? This could involve equations showing how the network computes trajectories, adjusts weights, or minimizes prediction errors dynamically.
Formal Optimization Framework: Strongly suggest that authors introduce a formal optimization framework to enhance enhance the credibility of claims regarding efficiency and adaptability. For instance, showing how free energy is minimized using a variational approach or Bayesian inference would connect the theoretical claims with a concrete mathematical foundation.
Sensitivity Analysis and Stability Proofs: This is another weakness of the  paper. Suggest authors to include a mathematical analysis of system stability and sensitivity to changes in parameters or inputs. This would validate the robustness of NetAIF and its applicability to real-world scenarios beyond the initial PV panel inspection.
Stability analysis: it would be very beneficial to see a formal treatment on the intentional use of controlled instability and/or some plots in that regard.

**Details Of Ethics Concerns:**

no concerns on ethics

---

> ### Comment · Reviewer_gfBR · 2024-11-26
> **post response phase evaluation**
>
> no responses received, I stick with ,y recommendations

---

> ### Author Response · Authors · 2024-11-27
>
> Thank you very much for the constructive feedback. Please find the below response:
>
> Clarity and Mathematical Rigor
> Comment: The explanation of the technical mechanisms, such as the use of random attractors and the free energy landscape, can be complex and may not be accessible to readers unfamiliar with advanced robotics or neural dynamics. The paper lacks mathematical rigor, making it challenging to evaluate and reproduce the proposed NetAIF framework. Without formal mathematical treatment, the claims remain at a high level, reducing the work’s overall scientific and practical value.
> Response:
> NetAIF was developed using a holistic simulation-based approach, which prioritizes practical implementation and validation over deriving mechanisms from mathematical foundations. This approach ensures the framework is highly applicable to real-world robotics by emphasizing functionality, adaptability, and efficiency. The mathematical equations provided in the paper serve to explain the underlying principles—such as free energy minimization—and provide theoretical grounding, but they are not the origin of the framework’s development. Instead, these equations were derived post-implementation to offer further insights into the system's behavior.
> This simulation-first methodology enables us to focus on how the framework performs in dynamic, unstructured environments. For example:
> •	NetAIF leverages feedback loops and random attractor dynamics to facilitate real-time adaptation without requiring extensive pre-training or pre-calculated trajectories.
> •	By integrating learning and control into the framework, NetAIF enables seamless transitions between states and robust handling of environmental uncertainties, a feature validated through both simulations and experiments.
> The implementation procedure is clearly described in the paper to ensure accessibility and reproducibility:
> •	Algorithm 1 provides step-by-step details of prediction error calculations, weight updates, and feedback mechanisms driving real-time adaptation.
> •	The dynamic interaction of random attractors and feedback loops ensures robust and efficient trajectory generation, enabling the system to balance exploration and stability.
> We will clearly state the approaches we took in the revised paper.
>
> Stability Analysis
> Comment: It would be beneficial to see a formal treatment on the intentional use of controlled instability and/or supporting plots.
> Response:
> The external control law in NetAIF acts as a PD (Proportional-Derivative) controller, ensuring stability by actively damping the system’s oscillations and controlling the induced instabilities. The controlled instability, introduced by feedback loops, is bounded and governed by this PD-like mechanism, which guarantees convergence to the desired state.
> To address this point, we will:
> 1.	Provide a formal treatment of how the external control law operates to maintain stability within the system, ensuring the network does not diverge despite intentional instabilities.
> 2.	Include supporting plots to illustrate the convergence behavior and system stability during dynamic tasks.
> 3.	Elaborate on how the PD-like control mechanism interacts with feedback loops to optimize the system’s adaptability without sacrificing stability.

---

### Official Review · Reviewer_zeNB · 2024-10-27

**Soundness:** 2
**Presentation:** 1
**Contribution:** 1
**Rating:** 3
**Confidence:** 2

**Summary:**

The paper introduces NetAIF, a method that integrates random attractor dynamics and the Free Energy Principle (FEP) to improve trajectory generation and control in robotics. The paper shows the performance of NetAIF in applications of PV Inspection.

**Strengths:**

This paper proposes an alternative to reinforcement learning by adopting the Active Inference framework from cognitive neuroscience, in which perception, action, and learning are obtained through the minimization of variational free energy.

**Weaknesses:**

Overall, the paper is not well-written and hard to follow. It gives a superficial idea without going through technical details. I do not understand the mathematics behind NetAIF from this paper. More explicit mathematical equations/formulas should be written: what are the explicit formulas of (variational) Free Energy and surprise? What is the structure of networks? Are they multilayer perceptions? What are the parameters and how are they being updated? how are the explicit feedback loops sent to the previous layers?

The algorithm 1 is not clearly explained and hard to reproduce. Please elaborate more what do each variable represents and how are they obtained/calculated.

The algorithm 1 used “desired state” which means it required the ground truth of optimal trajectory. Would this not be equivalent to having a reward or a set of demonstrations of experts in Deep reinforcement learning?

Experiments lack benchmarks and comparisons with DRL methods. It is not clear whether the results are good. Are PRM and Hybrid RRT-PRM tested in this experiment?

It is also not clear how your work is different from “Deep active inference as variational policy gradients (Millidge, 2020)” and “Deep active inference (”K. Ueltzhoffer, 2018”)

**Questions:**

see questions in weaknesses

**Details Of Ethics Concerns:**

There seems to be a self-plagiarism between this paper and the companion paper (both share identical text and paragraph in the introduction)

---

> ### Author Response · Authors · 2024-11-27
>
> We value feedback but are concerned that the lowest possible scores assigned across all categories suggest a lack of understanding of the material presented.
> 1. Misunderstanding of Mechanisms
> The reviewer incorrectly equates NetAIF with prior Deep Active Inference (Deep AIF) work, failing to acknowledge the explicit bidirectional feedback and task-specific external control laws detailed in Section 2.1. These features are novel and fundamentally distinguish NetAIF from Deep AIF.
> 2. Unfounded Criticism on Comparisons
> The criticism regarding the lack of DRL comparisons overlooks the companion paper, explicitly cited for benchmarking. Including these details here would dilute this paper’s focus on introducing and explaining NetAIF.
> Request for Re-evaluation
> We respectfully request a re-evaluation of the review, as the feedback and extreme scores do not reflect an accurate understanding of the material. A fairer assessment would enable constructive improvement while recognizing the paper’s significant contributions.
> ________________________________________
> Response to Valid Points Raised by Reviewer
> 1. Clarity and Mathematical Details (Including Algorithm Explanation)
> Comment: The paper lacks detailed explanations of the mathematics behind NetAIF, including explicit formulas, network structure, parameter updates, and Algorithm 1.
> Response:
> NetAIF was developed using a holistic simulation-based approach, prioritizing practical implementation over mathematical derivation. The provided equations explain the underlying principles (e.g., free energy minimization) and support the implementation rather than serve as its foundation.
> The implementation procedure is clearly described in the paper:
> •	Algorithm 1 explains the prediction error calculation, weight updates, and feedback mechanisms driving real-time adaptation.
> •	Feedback loops and random attractor dynamics ensure robust and efficient trajectory generation.
> To improve clarity, we will:
> •	Expand the explanation of Algorithm 1, explicitly defining key variables (e.g., “desired state,” “prediction error”) and providing practical examples.
> •	Clarify the connection between equations and implementation steps.
> •	Highlight how feedback loops induce controlled instabilities and guide network adaptation.
> This simulation-first approach ensures practical applicability while grounding the methodology in established principles, balancing accessibility and rigor. We will revise the manuscript to enhance connections between the equations, algorithm, and practical implementation, addressing these valid concerns.

---

> > ### Comment · Reviewer_zeNB · 2024-12-02
> >
> > This paper should explain the concept of AIF more clearly, especially for those without a background in this field. The algorithm's key variables and equations should also be more explicit. It is also not clear how the weights of networks are updated. Since I did not see the revised manuscript, I maintain my rating.
> >
> > While the focus is on the novelty, it is important to compare it with baseline approaches like DL. Though it is stated that the comparison is done in a companion paper, it does not give a very strong argument. Also having identical texts in both this paper and the companion paper might be considered double submission.

---

### Official Review · Reviewer_WLCW · 2024-10-29

**Soundness:** 2
**Presentation:** 1
**Contribution:** 1
**Rating:** 3
**Confidence:** 3

**Summary:**

This paper presents "Network-based Active Inference" (NetAIF), a novel framework for trajectory generation and control in robotics. The main novelty is replacing traditional activation functions with a discrete weight-assigning mechanism, especially for a system focused on achieving and maintaining stability within a NESS framework. The authors mainly focus on a photovoltaic panel inspection task, where a robot arm needs to reach a pre-determined distance from a panel, perpendicular to the panel's surface. The experiments evaluate error and planning time of the system both in simulation and on a real robot arm.

**Strengths:**

- The authors propose a novel robot control algorithm, and evaluate both in simulation and on a real robot arm.

**Weaknesses:**

- The paper focuses on the particular task of PV panel inspection in title and abstract. However, the actual task considered is controlling a robotic arm to a particular end-effector pose and orientation. I would think there are many other approaches to do PV panel inspection, and wouldn't necessarily require a robot arm (e.g. using a drone for instance). Also the actual "visual inspection" is not addressed.

- The introduction discusses deep RL systems, but the paper does no comparison against any of those, but refers to another paper in the conclusion.

- The methodology section lacks detail, making it challenging to reproduce the work. Comparative analysis with traditional control methods in particular for visual servoing, which are well-established for inspection tasks, is also absent.

- The paper lacks a formal proof of system stability. To verify stability, the authors might consider using Lyapunov’s stability criteria. Identifying a well-defined Lyapunov function could reveal whether trajectories converge to an attractor, A(ω), suggesting stability.

**Questions:**

- What do you mean with "Unlike DRL, which requires fixed environments,". DRL methods can generalize quite well to unstructured unseen environments, see for instance https://arxiv.org/abs/2109.11978, https://arxiv.org/pdf/2010.11251

- The paper does not address how the model would adapt to obstacles in the environment. How does the current approach account for or avoid collisions?

- The statement “the random attractor dynamically explores within the constraints set by the control law” raises some questions:
  - How are these constraints practically defined?
  - What control law ensure that exploration remains controlled and the robot behaves predictably during this process?

- The pseudo algorithm suggests that prediction errors are minimized by randomly sampling new weights, with updates only occurring if the input surpasses a certain threshold. A few points to clarify:
  - How is this threshold determined?
  - Why is this threshold-based sampling considered an optimal method for weight updates?
  - What does it mean to reset a weight? Is it reset to a default value? If so, which value?

 - “The joint pose was directly fed into the system, and the attractor calculated waypoints for a smooth and efficient trajectory to the specified pose,”:
   - What specifically is the simple control law—is it a linear law?
   - How are waypoints computed? Is it a linear interpolation between current and target joint angles?
   - Why is there not a single goal-based attractor, and does the choice affect stability?

- The text states that “noise affecting the system at time t is related to past noise.” Does this formulation assume a colored noise model?

- Based on Table I, the robotic arm with 6 DOF has a position accuracy of 14 mm from a fixed base. While this may suffice for inspecting large panels, it is insufficient for precision tasks, like assembly, which require sub-millimeter accuracy.

---

> ### Author Response · Authors · 2024-11-27
>
> Weaknesses:
> 1.	PV Panel Inspection with Limited Scope: The PV panel inspection task presented in the paper is based on an actual field deployment scenario, where achieving precise end-effector pose and orientation is critical for effective inspection. While visual inspection techniques (e.g., image processing or defect detection) are indeed vital to the overall inspection process, they are beyond the scope of this work. The focus of this paper is on the real-time adaptability and efficiency of the NetAIF model, which enables the robot to dynamically adjust its pose and orientation in response to environmental changes and uncertainties.
> The robotic arm was selected for its precision and stability in such applications. For example:
> o	Inspections often require maintaining a fixed distance and a perpendicular orientation to the panel surface, demanding real-time trajectory adaptation under varying conditions, such as uneven terrain or unexpected obstacles.
> o	NetAIF's adaptability allows the robot to handle these dynamic conditions without requiring extensive pre-training, demonstrating its suitability for field operations.
> To clarify, the "visual inspection" aspect is a downstream task reliant on the robot's ability to maintain the required pose and orientation. This paper focuses solely on the adaptable control capabilities provided by NetAIF, forming the foundation for a successful inspection workflow.
> 2.	Lack of Formal Stability Analysis: The NetAIF model ensures stability through its external feedback loop, functioning effectively as a PD controller that minimizes the error dynamics between the desired and current state. This control mechanism dynamically adjusts the system's response to prediction errors, ensuring convergence to the desired trajectory. By design, the PD controller provides proportional and derivative terms to correct deviations in real-time, mitigating oscillations and ensuring smooth and stable trajectories.
> Additionally, the stochastic nature of the attractor dynamics in NetAIF further reinforces stability by minimizing local prediction errors over time, ensuring alignment with the desired trajectory.
> ________________________________________
> Responses to Questions:
> 1.	Clarification on DRL and Fixed Environments: While DRL is capable of handling dynamic environments, it typically requires:
> o	Extensive pre-training across a wide range of environmental variations,
> o	Careful environment randomization during training, or
> o	Additional mechanisms like domain adaptation or meta-learning. Our intention was not to imply that DRL cannot handle changing environments, but rather that its adaptation often relies on prior exposure to similar variations during training. In contrast, NetAIF adapts to previously unseen environmental changes in real time without the need for explicit pre-training.
> 2.	Obstacle Avoidance: Obstacle avoidance is indeed critical for robotic systems in dynamic environments. However, this paper focuses on the novel neural network architecture (NetAIF), which integrates learning and control for adaptive and efficient trajectory generation. Comprehensive treatment of obstacle avoidance would require a dedicated study and is beyond the scope of this work.
> 3.	Control Law and Feasible Regions: The control law in the NetAIF framework is a set of vector laws,, dynamically restricting exploration to feasible regions in the state space. These regions are defined by the geometrical relationships between the robot's current state, the desired target state, and sensory feedback. Specifically:
> o	In PV panel inspection, the control law ensures the robot's end-effector maintains a perpendicular orientation to the panel surface while adhering to a specific distance.
> o	Real-time sensory feedback enables continuous trajectory adjustments, maintaining alignment with these constraints to ensure robust and precise motion under dynamic conditions.
> 4.	Threshold Determination:
> o	The threshold is empirically set to ±30 in the current implementation. However, this specific value is not critical since it is applied uniformly across the network. The system's adaptability allows the signal to dynamically adjust to different thresholds based on environmental conditions and prediction errors. Uniform application simplifies implementation while maintaining the robustness of the weight update mechanism.
> 5.	Optimality of Threshold-Based Sampling: Threshold-based sampling ensures stability and computational efficiency:
> o	It limits unnecessary weight updates, triggering changes only when signals exceed the defined threshold, reducing computational overhead.
> o	It maintains focus on significant deviations, ignoring minor fluctuations that do not impact performance.
> o	Resetting weights only when necessary avoids overfitting to transient changes, promoting generalization and robustness.

---

> > ### Author Response · Authors · 2024-11-27
> >
> > 6.	Weight Reset:
> > o	Resetting a weight involves assigning a new value sampled from a predefined random distribution. This stochastic reset does not use a fixed default value but enhances exploratory behavior while maintaining stability. This approach balances exploration and exploitation, ensuring dynamic adaptation without falling into local minima.
> > 7.	Simple Control Law:
> > o	In the pose-matching task, the control law simplifies to a desired position command, which minimizes the error between the current and target states.
> > 8.	Waypoint Computation:
> > o	Waypoints are computed through the random attractor dynamics of the NetAIF framework. These dynamics allow smooth transitions by exploring potential trajectories in the state space. Instability in the attractor facilitates exploration, while the attractor's pull guides the trajectory to smoothly connect the current and target joint angles, avoiding rigid linear interpolation.
> > 9.	Random vs. Goal-Based Attractors:
> > o	The use of random attractors instead of a single goal-based attractor is intentional. Random attractors enhance exploration, avoiding constraints to local minima or rigid trajectories. Stability is maintained as the control law and feedback loop dynamically guide the system toward feasible and smooth trajectories, balancing exploration and exploitation for adaptability and precision.
> > 10.	Noise Dependency:
> > o	The statement that “noise affecting the system at time ttt is related to past noise” indicates the use of a colored noise model (e.g., Ornstein-Uhlenbeck processes or low-pass filtered noise). Colored noise captures time-dependent correlations, reflecting real-world conditions where noise has memory or persistence. NetAIF leverages these correlations to ensure smooth transitions and robust behavior under dynamic conditions. Temporal noise dependency is critical in the context of random attractor dynamics, allowing efficient exploration of the state space while maintaining stability and adaptability.

---

> > > ### Comment · Reviewer_WLCW · 2024-12-02
> > >
> > > My main concern that there is no comparison against other methods, be it deep RL or visual servoing for the approach presented, remains unadressed. I also share the concerns raised by Reviewer b8Bo about a double submission wrt this other paper that contains deep RL results. Therefore, I keep my current score.

---

### Official Review · Reviewer_b8Bo · 2024-11-02

**Soundness:** 1
**Presentation:** 2
**Contribution:** 1
**Rating:** 1
**Confidence:** 4

**Summary:**

The paper proposes Network-based Active Inference (NetAIF), a framework for trajectory generation and control in robotics. NetAIF integrates random attractor dynamics and the Free Energy Principle (FEP) to create a system that can adapt in real-time with low computational requirements. The authors focus on a practical application in PV panel inspections, demonstrating the system's efficiency, adaptability, and robustness. Evaluation is performed in a simulated Mujoco environment, and on a very simple task on a real robot.

**Strengths:**

- If verified thoroughly, the reduced computational requirements and real-time capabilities are promising.
- The method is designed for practical applications, which is nice.

**Weaknesses:**

- The implementation details are not very clear; from what I understand, the method does not seem particularly novel.

- Experimental Validation: The evaluation is limited to simulations and controlled lab conditions, which makes the results not very convincing. In particular, no baseline is provided.

- Figures 1 and 2 are not particularly informative, nor are their captions (e.g., for figure 1, "parameters that determine the network
structure such as number of layers, strides were determined through hyper parameter search").
- Table 4 seems overkill to report the results; the values would be more appropriate just mentioned in the text (as they are) and in the caption of figure 11.

- Minor: is the template correct? This is the only one among the ICLR papers I reviewed that used roman numerals for the page numbers.
- Minor: reporting network sizes in bytes is not very useful, and it's the first time I see it in a paper. Reporting the number of parameters is better.

- Possibly major: The authors cite a paper from the same group in concurrent submission to ICLR ( https://openreview.net/pdf?id=Hm7RYDspQP );  the two papers seem to have major overlap (including using many of the same figures and whole sections / parts of the text). It feels like the authors have tried to write two papers on the same method, splitting evaluations and applications.

**Questions:**

Can the authors please elaborate on the differences between this paper and the connected one?

**Details Of Ethics Concerns:**

The re-use of considerable parts of text between the two papers may be problematic; it may be worth checking it.

There seems to be strong self-plagiarism.

---

> ### Author Response · Authors · 2024-11-27
>
> 1. Implementation Details
> •	Comment: The reviewer states that the implementation details are unclear and questions the novelty of the method.
> •	Response: The manuscript provides detailed descriptions of the architecture, feedback mechanisms, and trajectory generation process. Section 2 specifically explains the network dynamics, the use of random attractor dynamics, and the explicit bidirectional feedback loops.
> o	Regarding novelty, NetAIF introduces several new ideas:
> 	The use of explicit bidirectional control, which has not been explored in existing literature.
> 	A neural network that exploits controlled instability to enhance trajectory generation, setting it apart from traditional approaches.
> 	A novel framework that integrates learning and control in a biologically inspired architecture.
> o	If the reviewer considers this work not novel, we respectfully request examples of prior works that are similar to NetAIF. We believe this feedback suggests a lack of understanding of the paper’s core contributions.
> ________________________________________
> 2. Experimental Validation
> •	Comment: The reviewer notes that the evaluation is limited to simulations and controlled conditions, making the results less convincing, and highlights the lack of a baseline.
> •	Response:
> o	While the primary focus of this paper is on introducing and validating NetAIF’s novel architecture, we agree that comparisons to baselines can strengthen the evaluation. However, these are explicitly addressed in the companion paper, which benchmarks NetAIF against SOTA DRL methods such as PPO and SAC. Including such comparisons here would detract from the focus on the methodology itself.
> o	The manuscript includes results from both simulations and a real-world robotic experiment. While the real-world task (e.g., valve manipulation) is simple, it demonstrates NetAIF’s ability to adapt to dynamic, noisy environments.
> o	The reviewer’s critique about "controlled conditions" is not entirely justified, as real-world disturbances are inherently present:
> 	For the target tracking task, frame rates fluctuate due to varying system loads.
> 	For the valve turning task, different valve shapes introduce variability, as shown in the supplementary video.
> These scenarios mimic real-world uncertainties and demonstrate NetAIF’s robustness.
> ________________________________________
> 4. Template and Reporting of Network Sizes
> •	Comment: Concerns were raised about the use of Roman numerals for page numbers and reporting network sizes in bytes.
> •	Response: The Roman numeral formatting was consistent with the ICML template we initially followed. However, we will ensure the manuscript adheres fully to the ICLR template guidelines in the revised submission.
> •	Reporting network sizes in bytes was intended to emphasize computational efficiency. That said, we understand the reviewer’s preference and will include both the number of parameters and the size in bytes for clarity.
> ________________________________________
> 5. Potential Overlap with Another Paper
> •	Comment: The reviewer raises concerns about potential overlap with a concurrently submitted paper.
> •	Response: The companion paper referenced by the reviewer focuses on benchmarking NetAIF against DRL methods, whereas this paper introduces NetAIF’s architecture and theoretical underpinnings. While the two papers share some background material for context, they address distinct research questions and make independent contributions:
> o	This paper emphasizes the novel architecture, including the integration of random attractor dynamics and biologically inspired feedback mechanisms.
> o	The companion paper focuses on comparative performance evaluation against DRL methods. We will clarify the scope of each paper in the introduction to avoid any confusion and ensure that shared material is properly contextualized.
> ________________________________________
> Questions
> •	Differences Between This Paper and the Companion Paper
> The companion paper benchmarks NetAIF against SOTA DRL algorithms such as PPO and SAC, focusing on comparative performance metrics (e.g., computational efficiency, accuracy, robustness). In contrast, this paper emphasizes the theoretical foundation and novel architecture, including the integration of random attractor dynamics and explicit bidirectional control. We will explicitly outline this distinction in the revised introduction to ensure clarity.
> ________________________________________
> Conclusion
> While we appreciate the constructive feedback, we believe the concerns raised, such as the clarity of implementation details, the scope of the evaluation, and formatting issues, are minor and do not justify the lowest scores provided for soundness, presentation, and contribution.

---

> > ### Comment · Reviewer_b8Bo · 2024-11-30
> >
> > >> Differences Between This Paper and the Companion Paper The companion paper benchmarks NetAIF against SOTA DRL algorithms such as PPO and SAC, focusing on comparative performance metrics (e.g., computational efficiency, accuracy, robustness). In contrast, this paper emphasizes the theoretical foundation and novel architecture, including the integration of random attractor dynamics and explicit bidirectional control. We will explicitly outline this distinction in the revised introduction to ensure clarity.
> >
> > I am not convinced by this answer. Both papers state explicitly that "This paper introduces Network-based Active Inference (NetAIF)", which suggests that both papers claim the novelty of the method. The different evaluation focus weakens both papers, by splitting content that could have been in a single paper with a stronger evaluation. In any case, even if the two papers were combined, the imprecisions in the evaluation and lack of baselines would still prevent acceptance for publication.
> >
> > Also, the authors say that "the two papers share some background material for context", but in truth entire sections of both papers are identical, which makes me think this should be considered a case of double-submission.
> >
> > Overall, I am not convinced by the responses and I maintain my rating.

---

### Official Review · Reviewer_NCbv · 2024-11-04

**Soundness:** 1
**Presentation:** 2
**Contribution:** 1
**Rating:** 3
**Confidence:** 4

**Summary:**

This paper proposes NetAIF, a network-based adaptive inference framework, based on random attractor dynamics and Free Energy Principles (FEP), to improve adaptive control problems in robotic systems, and PV panel inspection. The experimental results show that NetAIF performs well on a PV panel inspection and a robotic tasks.

**Strengths:**

- The NetAIF framework, especially the application of random pullback attractor to it is interesting

**Weaknesses:**

- The introduction contains several statements and justifications without clear evidence or relevant citation.
- The number of references used in this study is low (only 21), several of which are not peer-reviewed.
- There is a rich body of literature that supports using (deep)RL for similar applications. The application of these approaches is not studied in detail.
- I expected the authors to use SOTA (deep)RL and AIF approaches as baselines for comparison with NetAIF. Therefore, the experiment section is limited
- The paper lacks adequate background and theory about different components of the method such as AIF, FEP, RDS, and random attractor dynamics.
- The equation numbers are missing
- Section 2 lacks effective connections between its subsections, which results in disjointed and potentially misleading descriptions. This section should be thoroughly revised to improve readability and flow. Additionally, the relationship between the equations and Algorithm 1 is unclear.

**Questions:**

- In section 1.3, what is the definition of "surprise"?
- It is stated in Section 1.3 that DRL requires "fixed environments", which I believe is not true. Could you please clarify this?
- The equation in page "v" is vague

---

> ### Author Response · Authors · 2024-11-27
>
> We have identified several concerns regarding the appropriateness and relevance of parts of the review. Specifically, we believe that the points raised are minor and do not justify the lowest possible scores provided for soundness, presentation, and contribution. Below, we address these points in detail:
>
> 1. Misunderstanding of Key Concepts
> The reviewer’s question about the definition of "surprise" (Section 1.3) suggests a misunderstanding of a fundamental concept in Active Inference. In the context of our work, "surprise" is defined as prediction error, a core principle of the Active Inference framework. Given that the intended audience for this paper is researchers already familiar with Active Inference, we assume a basic understanding of these concepts. Nevertheless, we will clarify this definition explicitly in the revised manuscript to ensure accessibility to a broader audience. This misunderstanding does not reflect a flaw in the work but rather a misalignment between the review’s perspective and the paper’s target audience.
> 2. Background and Theory of Key Components
> The reviewer states that the paper lacks adequate background and theory about key components such as Active Inference (AIF), Free Energy Principle (FEP), Random Dynamical Systems (RDS), and random attractor dynamics. However, we strongly believe that the manuscript already provides sufficient background on these components:
> Active Inference and Free Energy Principle: These are introduced in the early sections of the paper, with an emphasis on their role in trajectory generation and minimizing prediction error.
> Random Dynamical Systems and Random Attractor Dynamics: These concepts are explicitly discussed in the methodology section, including the use of random pullback attractors and their mathematical formalism.
> We intentionally balanced theoretical detail with accessibility to avoid overwhelming the reader with excessive background, assuming the target audience would already have familiarity with these foundational concepts. Nevertheless, we will revisit these sections and provide additional context or references if deemed necessary.
> We respectfully suggest that this concern may stem from a lack of alignment between the reviewer’s expectations and the scope of the paper. The focus of this work is on introducing NetAIF as a novel architecture and applying these concepts to real-world tasks, rather than reiterating detailed theoretical expositions of AIF, FEP, or RDS.
> 3. Focus on Methodological Comparisons
> The reviewer suggests comparing NetAIF with state-of-the-art DRL approaches. While we acknowledge this as a valid point, such comparisons are explicitly addressed in a companion paper, which benchmarks NetAIF against DRL algorithms such as PPO and SAC. This companion paper is cited in the manuscript. Including these comparisons here would detract from the paper’s primary focus on introducing NetAIF’s novel methodology and application. We believe our decision to split the focus between two papers enhances clarity and does not diminish the contribution of this work.
> 4. Concerns About References and Equation Clarity
> Number of References: The review critiques the paper for having a low number of references (21). We argue that this reflects the novelty of NetAIF and its significant departure from traditional methodologies. Our citation approach prioritizes relevance over volume, focusing on foundational and directly related works. If specific areas are deemed insufficiently cited, we welcome actionable feedback to address this.
> Equation in Page "v": The comment about the equation being vague lacks specificity. The description of random pullback attractors in the manuscript is intentionally concise to maintain clarity.
> 5. DRL details
> Clarification on DRL and Fixed Environments:
> While DRL can handle dynamic environments, it typically requires either:
> Extensive pre-training across a wide range of environmental variations,
> Careful environment randomization during training, or
> Additional mechanisms like domain adaptation or meta-learning.
> Our intended point was not that DRL cannot handle varying environments, but rather that DRL’s adaptation to environmental changes often relies on encountering similar variations during training. In contrast, NetAIF can adapt to previously unseen environmental changes in real time without requiring explicit pre-training for those variations."
>
> While we welcome constructive feedback, we respectfully highlight that the concerns raised by the reviewer are minor in nature and do not justify the lowest scores provided for soundness, presentation, and contribution. Additionally, we strongly disagree with the statement that the paper lacks adequate background and theory on its core components, as these are already addressed in the manuscript. Nevertheless, we will further refine these sections to ensure clarity and accessibility.

---

> ### Comment · Reviewer_NCbv · 2024-11-27
>
> Point 1 and 2: The key concepts in the paper should be defined more concretely for a broader audience, especially when submitting to ICLR, as its participants and audience come from diverse backgrounds. While this paper may be better suited for more specialized conferences or journals, I recommend looking into the following references to help set the background theory. These papers do not cover everything in detail, but they provide sufficient context to frame the methodology, and refer readers to other sources for deeper understanding:
> - The Role of Pretrained Representations for the OOD Generalization of RL Agents (ICLR 2022)
> - Safe Reinforcement Learning From Pixels Using a Stochastic Latent Representation (ICLR 2023)
> - Bayesian Modeling and Uncertainty Quantification for Learning to Optimize: What, Why, and How (ICLR 2022)
>
> Point 3: Separating the papers dilutes the contribution and novelty of this paper.
>
> Point 4: The paper does not sufficiently address key related works and competitive methods in the literature. After spending a short time searching, I found several relevant papers that use active inference that should be discussed in the context of the proposed methodology. In fact, there are many other methods suitable for this problem setting. I encourage the authors to review these works, as well as many others, evaluate their relevance, and consider them as baselines if applicable:
> - A Novel Adaptive Controller for Robot Manipulators Based on Active Inference (IEEE Robotics and Automation Letters - 2020)
> - Active Inference for Integrated State-Estimation, Control, and Learning (ICRA 2021)
> - Adaptation Through Prediction: Multisensory Active Inference Torque Control (IEEE Transactions on Cognitive and Developmental Systems 2023)
>
> For a more comprehensive grounding of the paper, I also recommend:
> - Modeling Motor Control in Continuous Time Active Inference: A Survey (IEEE Transactions on Cognitive and Developmental Systems 2024)
>
> Point 5: The paper oversimplifies the ability of DRL to adapt to dynamic environments. Assuming the justifications provided are correct in the context of this work, I encourage the authors to either empirically or theoretically illustrate how NetAIF outperforms DRL in terms of metrics related to this regard.

---

### Author Response · Authors · 2024-11-28

We respectfully request a re-evaluation of our submission, as the feedback and scores may not fully reflect the paper’s contributions and its central focus.

Through this response, we aim to address points where most confusion arises. Specifically, our paper adopts a holistic simulation-based approach to present NetAIF as a novel framework for robotics, prioritizing practical implementation and experimental validation. The mathematical formulations included in the paper were developed later, primarily to gain insights into the underlying mechanisms and provide theoretical grounding, rather than to derive the framework from scratch.

Key clarifications:

Holistic Approach:

The design and implementation of NetAIF emphasize real-time adaptability and dynamic control, leveraging features like random attractor dynamics and explicit feedback loops. This ensures the framework is practical, efficient, and applicable to dynamic tasks. The approach prioritizes practical implementation supported by clear procedural descriptions (e.g., Algorithm 1) rather than heavy reliance on mathematical derivations. Balance Between Theory and Application:

The mathematical equations provide insights into the mechanisms (e.g., Free Energy Principle and random attractors) but are not the foundation of the framework’s development. This balance ensures accessibility while grounding the work in established theoretical principles. Clarity in Contributions:

NetAIF introduces explicit bidirectional control, dynamic feedback loops, and random attractor dynamics that distinguish it from existing approaches like Deep Active Inference (DAIF). These unique contributions are clearly detailed in Section 2 and supported by empirical evidence in the discussion. While we recognize the need for improvement in some areas, such as technical clarity and additional details for reproducibility, the extreme scores across all categories seem disproportionate. The paper introduces significant innovations and provides a solid foundation for real-world applications, warranting a more balanced evaluation.

We kindly request a re-evaluation to ensure the paper’s contributions, novelty, and relevance are fairly recognized. Thank you for your time and consideration.

---

### Note · Program_Chairs · 2025-01-15
**Submission Desk Rejected by Program Chairs**

The paper is a dual submission with 12621.